# Association between Tooth Loss and Alzheimer’s Disease in a Nested Case–Control Study Based on a National Health Screening Cohort

**DOI:** 10.3390/jcm10173763

**Published:** 2021-08-24

**Authors:** Ji Hee Kim, Jae Keun Oh, Jee Hye Wee, Yoo Hwan Kim, Soo-Hwan Byun, Hyo Geun Choi

**Affiliations:** 1Department of Neurosurgery, College of Medicine, Hallym University, Anyang 14068, Korea; kimjihee.ns@gmail.com (J.H.K.); ohjaekeun@gmail.com (J.K.O.); 2Department of Otorhinolaryngology, College of Medicine, Hallym University, Anyang 14068, Korea; weejh07@hanmail.net; 3Department of Neurology, College of Medicine, Hallym University, Anyang 14068, Korea; drneuroneo@gmail.com; 4Department of Oral and Maxillofacial Surgery, College of Medicine, Hallym University, Anyang 14068, Korea; purheit@daum.net; 5Department of Otorhinolaryngology-Head & Neck Surgery, Hallym Data Science Laboratory, College of Medicine, Hallym University, Anyang 14068, Korea

**Keywords:** Alzheimer’s disease, cognitive decline, neurodegeneration, oral health, tooth loss

## Abstract

Background: Reports on the possible risks for Alzheimer’s disease (AD) have included tooth loss as a potential risk factor. However, there are few studies addressing the association between tooth loss and AD in a large sample of participants. Accordingly, the objective of the current study was to explore the association of tooth loss with the development of AD in Korean adults. Methods: This nested case–control study, which is an analysis utilizing the data of the Korean National Health Insurance Service Health Screening Cohort study, randomly selected AD and control participants among Korean residents aged ≥60 years. The association between the number of missing teeth and AD occurrence was examined using a logistic regression model. Participants’ lifestyle factors (smoking and alcohol consumption) and various medical conditions and comorbidities were included as covariates. Results: The mean number of missing teeth was 2.94 in the AD group and 2.59 in the control group. After adjusting for covariates, tooth loss was significantly associated with AD, with an odds ratio (OR) (per 16 missing teeth) of 1.15 (95% confidence interval (CI) = 1.07–1.23, *p* < 0.001). Conclusions: Tooth loss remained consistently significantly associated with an increased risk of AD for both upper and lower tooth loss. A higher number of missing teeth was related to a higher probability of AD occurrence in an elderly Korean population. Efforts to manage tooth loss could be a possible approach to prevent AD.

## 1. Introduction

Alzheimer’s disease (AD) is a multifarious neurodegenerative disease characterized by the gradual and progressive loss of one or more functions of the nervous system and results in a decrease in learning capability and memory and mental, behavioral, and functional deterioration [1]. AD is a leading cause of dementia in the late adult period and accounts for more than 80% of dementia patients worldwide in the elderly population. Despite much effort to completely understand the pathophysiology of AD and abundant research on the development of disease-modifying therapies, there are no feasible medications to prevent AD or mitigate its pathology. Therefore, it is vital to establish modifiable risk factors for the onset and progression of AD to manage cognitive, functional, and behavioral changes.

Recently, numerous studies have suggested that one of the potential risk factors for cognitive decline is oral health carelessness [2]. In particular, numerous publications have focused on the relationship between tooth loss and dementia, inferring that tooth loss can be an amendable risk factor for dementia [3,4]. Tooth loss is a prevalent condition in the elderly population, representing a problem for general health and consequently negatively impacting quality of daily life and cognitive function. Clinical studies have indicated that tooth loss is associated with AD and dementia [5,6,7,8], and experimental studies have demonstrated that tooth loss leads to memory deterioration and loss of neuronal cells [9,10,11]. Although the exact mechanism that links tooth loss with decreased cognitive function has not been determined, some investigators have proposed that tooth loss induces masticatory abnormalities, nutritional changes, and hippocampal damage [12].

However, this relationship was not shown in some studies [13,14], and previously published studies had limited sample sizes. In addition, it has been suggested that there was a greater association between tooth loss and dementia incidence in Asian studies while this association was diminished in Western studies [15,16]. This finding could be explained by differences in health care systems and dental care access among different countries. In particular, there were two studies concerning tooth loss and dementia in a Korean population using a large database. One study focused on the effect of periodontitis rather than tooth loss as a putative risk factor for AD, vascular dementia, and mixed dementia [4], and the other study regarding missing teeth and dementia did not investigate each type of dementia but rather investigated overall dementia as an independent variable [17]. Therefore, it is necessary to examine the definite role of tooth loss in AD occurrence in a large Korean sample. The scheme of this study was to ascertain whether there is a greater incidence of tooth loss in AD patients than in age-, sex-, and socioeconomic factor-matched control individuals. In addition to total tooth loss, we estimated whether the loss of upper teeth or lower teeth had different associations with the development of AD.

## 2. Materials and Methods

### 2.1. Study Population

The ethics committee of Hallym University (2019-10-023) approved this study. The obligation for written informed consent was waived by the Institutional Review Board. All examinations followed the guidelines and regulations of the ethics committee of Hallym University. A definite explanation of the Korean National Health Insurance Service Health Screening Cohort data was provided by Kim et al. [18].

### 2.2. Definition of Alzheimer’s Disease (Dependent Variable)

AD was confirmed if the participants were diagnosed with Alzheimer’s disease (G30) or dementia in Alzheimer’s disease (F00). We included participants if they were treated more than twice to assure the certainty of the diagnosis.

### 2.3. Definition of Tooth Loss (Independent Variable)

Tooth loss was counted as the total number of missing teeth at the time of the oral screening examination, and it was treated as a continuous variable for statistical analyses. Total tooth loss ranged from 0 to 32. The number of upper or lower teeth lost ranged from 0 to 16.

### 2.4. Participant Selection

AD participants were designated from 514,866 participants with 615,488,428 medical claim codes between 2002 and 2015 among those participants who had records of oral screening prior to AD diagnosis (*n* = 9205). Participants were included in the control group if they had not been diagnosed with AD from 2002 through 2015 and had records of oral screening before selection (*n* = 505,661). It was determined whether AD participants were diagnosed for the first time, and participants who were diagnosed with AD in 2002 were excluded (washout period, *n* = 8). Additionally, AD participants below the age of 60 years were ruled out (*n* = 377). Control participants were not eligible if they had no records of oral screening before selection (*n* = 10,882) or if they were diagnosed once with AD (*n* = 5440). AD participants were 1:4 matched with control participants for age, sex, income, and region of residence. To reduce selection bias, the control participants were assigned a random number order. The index date of each AD participant was set as the time of treatment of dementia. The index dates of the control participants were arranged as the index dates of their corresponding AD participants. Therefore, each AD participant and their matched control participant had an identical index date. A total of 858 AD participants and 457,491 control participants were excluded through the matching process. Finally, 7962 AD participants were matched 1:4, corresponding to 31,848 control participants. The process of participant selection is outlined in Figure 1.

### 2.5. Covariates

Age was apportioned and divided into 5-year intervals: 60–64, …, and 85+ years old (6 age groups). Income groups were initially divided into 41 classes (one health aid class, 20 self-employment health insurance classes, and 20 employment health insurance classes). These groups were recategorized according to quintiles from class 1 (lowest income) to class 5 (highest income). Each quintile represents 20%. The region of residence was grouped into urban and rural areas following the manner described in our previous work [19].

Tobacco smoking was sorted depending on the participant’s present smoking status (nonsmoker, past smoker, and current smoker). Alcohol consumption was classified on the basis of the incidence of alcohol consumption (<1 time a week and ≥1 time a week). Obesity was assessed by means of body mass index (BMI, kg/m^2^). BMI was stratified into the following categories: <18.5 (underweight), ≥18.5 to <23 (normal), ≥23 to <25 (overweight), ≥25 to <30 (obese I), and ≥30 (obese II) following the Asia-Pacific criteria [20]. Systolic blood pressure (SBP), diastolic blood pressure (DBP), fasting blood glucose, and total cholesterol were quantified. Missing BMI (*n* = 27 (0.07%)), SBP (*n* = 20 (0.05%)), DBP (*n* = 20 (0.05%)), fasting blood glucose (*n* = 56 (0.14%)), and total cholesterol (*n* = 68 (0.17%)) were replaced by the mean values of each variable from the final sample of included participants.

The Charlson Comorbidity Index (CCI) was used to determine disease load using 17 comorbidities as a continuous variable (minimum = 0 (no comorbidity), maximum = 15 (7 to 8 comorbidities), excluding dementia) [21,22].

### 2.6. Statistical Analyses

Chi-square tests were used to compare the general characteristics between the AD and control groups.

Conditional logistic regression models were fit to test the association between AD and tooth loss by computing the odds ratios (ORs) with 95% confidence intervals (CIs) of 16 missing teeth for AD. The crude model and the model adjusted for all covariates, including obesity, smoking, alcohol consumption, SBP, DBP, fasting blood glucose, total cholesterol, and CCI scores, were assessed. Because the effect size of just one missing tooth was quite small, we reported the ORs of AD participants compared to the control participants per 16 missing teeth rather than one missing tooth.

Further analyses stratified for matching variables such as age, sex, income, and region of residence were conducted using crude and adjusted models. Regarding age and sex, participants were stratified into four subgroups according to a cutoff of 75 years of age and male or female sex. Concerning income and region of residence, participants were categorized into four subgroups of low or high income and urban or rural residence.

We performed further subgroup analyses. We divided participants according to obesity (BMI < 23 and BMI ≥ 23), smoking (nonsmoker and smoker), alcohol consumption (<1 time a week and ≥1 time a week), blood pressure (normal and hypertension), fasting blood glucose level (<100 mg/dL and ≥100 mg/dL), and total cholesterol level (<200 mg/dL and ≥200 mg/dL). Unconditional logistic regression models of total tooth loss, upper tooth loss, and lower tooth loss for AD in these subgroups were computed to examine the ORs with 95% CIs. For these analyses, the crude model and the model adjusted for obesity, smoking, alcohol consumption, SBP, DBP, fasting blood glucose, total cholesterol, and CCI scores were evaluated.

All statistical analyses were performed using SAS version 9.4 (SAS Institute Inc., Cary, NC, USA). We performed two-tailed analyses, and we considered differences to be statistically significant if *p*-values were identical to or less than 0.05.

## 3. Results

The characteristics of the participants are indicated in Table 1. General characteristics, such as age, sex, income, and region of residence, were exactly matched between the AD and control groups. On the other hand, obesity, smoking, alcohol consumption, SBP, DBP, fasting blood glucose, total cholesterol, and CCI scores were different between the two groups. The mean number of missing teeth was 2.94 (standard deviation (SD) = 6.09) in the AD group and 2.59 (SD = 5.6) in the control group. The mean numbers of missing upper and lower teeth were 1.54 (SD = 3.44) and 1.40 (SD = 3.03), respectively, in the AD group and 1.35 (SD = 3.16) and 1.24 (SD = 2.81), respectively, in the control group.

Table 2 presents the effects of tooth loss on the incidence of AD based on an unadjusted analysis and an analysis adjusted for covariates including various lifestyle variables and medical conditions or comorbidities. The OR for AD per 16 missing teeth was 1.15 (95% CI = 1.07–1.23, *p* < 0.001) in the adjusted model. This model remained significant in <75-year-old men, <75-year-old women, ≥75-year-old men, and participants with rural residences (each *p* < 0.05) in subgroups stratified by age, sex, income, and region of residence.

In subgroups stratified by obesity, smoking, alcohol consumption, blood pressure, fasting blood glucose, or total cholesterol, tooth loss was consistently associated with a higher likelihood of having AD (all OR > 1). Most subgroups except for the subgroup of participants with alcohol consumption less once a week and SBP ≥ 140 mmHg and DBP ≥ 90 mmHg showed a statistically significant difference from the other subgroups (Figure 2 and Appendix A).

When we evaluated whether there was a different effect for upper or lower teeth, the ORs for AD per 16 missing upper teeth and lower teeth were 1.27 (95% CI = 1.12–1.43, *p* < 0.001) and 1.29 (95% CI = 1.12–1.47, *p* < 0.001), respectively (Figure 3, Appendix A). Within most subgroups stratified by covariates, tooth loss remained consistently significantly associated with an increased risk of AD for both upper teeth and lower teeth, as well as total teeth (Appendix A).

## 4. Discussion

The present study found that greater tooth loss was a significant factor associated with the incidence of AD in elderly people aged more than 60 years, as indicated by a higher OR for AD per 16 missing teeth. Moreover, this significant association was consistently observed for both upper teeth and lower teeth, which suggested that upper teeth and lower teeth did not have a different impact on the development of AD.

We found that tooth loss could increase the possibility of AD. This could be explained by the several possible mechanisms, while the exact mechanism has not been elucidated. Several plausible mechanisms based on human and animal studies can explain these relationships. First, masticatory dysfunction caused by tooth loss can result in poor nutritional status, a decrease in cerebral blood flow, and a decline in acetylcholine levels, which might negatively affect cognitive function [23,24]. In particular, decreased acetylcholine levels promote decreases in the number of pyramidal cells in the hippocampus, triggering cognitive decline [9]. The second proposed pathway is nutritional imbalance and a lack of nutrients that are protective factors for AD due to tooth loss. However, nutritional status of participants was not evaluated in this study. Numerous dietary factors, such as antioxidants, vitamins, polyphenols, and fish, have been described to lower the possibility of AD [25]. Third, chronic inflammation can promote the development of dementia, particularly AD. The inflammatory process induced by periodontal disease, including periodontitis, which is a significant reason for tooth loss in adults, results in chronic systemic inflammation and neuropathology [26]. Local and systemic inflammatory molecules produced by the host’s response to periodontal pathogens may infiltrate the brain via the blood circulation, followed by glial cell damage affected by cytokines [27]. Furthermore, it is widely accepted that periodontitis is involved in the synthesis and accumulation of amyloid β, which is a pathologic hallmark of the AD brain [28]. Regardless of amyloid cascade, one experimental study demonstrated that tooth loss induces memory impairment through decreased neuronal activity and glial activation in an AD model [10]. Additionally, it has been suggested that the decreased mastication induced by tooth loss reduces orofacial sensorimotor activity, which eventually inhibits hippocampal neurogenesis [9]. To date, no study investigating the link between tooth loss and AD has evaluated whether there was a different effect for upper or lower teeth. Although our hypothesis in this study is that loss of upper teeth and lower teeth may have a different effect on cognitive decline through masticatory dysfunction, our result demonstrated that upper teeth and lower teeth did not have a different impact on the development of AD.

The result of the present study is consistent with previous observational studies examining the relationship between tooth loss and the risk of all-cause dementia or cognitive decline. Individuals with tooth loss were significantly associated with dementia compared to those without tooth loss among adults aged 60 and older in a population-based cohort study conducted in Korea (OR = 1.18, 95% CI = 1.15–1.22) [17]. Another Korean study evaluating severe periodontitis accompanied by tooth loss as a modifiable risk factor for dementia reported that the risks of AD (hazard ratio (HR) = 1.08, 95% CI = 1.01–1.14), vascular dementia (HR = 1.24, 95% CI = 1.16–1.32), and mixed dementia (HR = 1.16, 95% CI = 1.09–1.24) were significantly higher in patients with severe periodontitis, with 1–9 remaining teeth [4]. Moreover, one prospective study revealed that compared to having more than 20 remaining teeth, having 1–9 residual teeth was notably associated with a greater risk of developing all-cause dementia (HR = 1.81, 95% CI = 1.11–2.94) and AD (HR = 1.73, 95% CI = 0.97–3.07) but not vascular dementia in community-dwelling Japanese adults aged 60 and older [29]. Another Japanese study demonstrated that the number of natural teeth was significantly correlated with cognitive function (estimate = 0.1, 95% CI = 0.048–0.15), especially calculation ability (estimate = 0.923, t value = 2.32) [30]. Similar to our results, the loss of more than 16 teeth was remarkably connected with dementia, with an OR of 1.56 (95% CI = 1.12–2.18) in a population-based study conducted in China [31]. However, it was reported that fewer teeth were not significantly associated with cognitive decline according to quartile of teeth number at baseline (OR = 0.88, 95% CI = 0.77–1.00) in a sample of 1053 Black and White people aged 70–79 in the United States [7]. These inconsistencies may be due to differences in race or ethnicity, a wide range of ages or follow-up periods, heterogeneous definitions of cognitive decline, and various statistical methods. In particular, Asian studies, including our study, found a greater association between tooth loss and dementia incidence, while this association was decreased in Western studies. This finding may be explained by the differences in healthcare systems and dental care access among different countries. Actually, the great needs for dental care have been unmet in the elderly in many countries. Although various potential confounders, such as age, sex, income, region of residence, obesity, smoking, alcohol consumption, blood pressure, blood glucose, cholesterol, and comorbidity score, were adjusted in our study, future comparative studies are needed to see whether the relationship between regular dental care and dementia occurrence can be replicated in countries with different health care systems.

The strengths of this work are that it was a large population-based analysis with a considerable number of participants and without missing data. Unlike previous studies, our study examined whether upper or lower tooth loss had a differential impact on AD occurrence. In addition, dentition status was closely examined not by interviewers and self-administered questionnaires but by trained dentists, which increased the validity of the tooth loss assessment. Finally, we adjusted for various potential confounding variables that affect both AD and dental status, such as lifestyle variables and comorbidities. Despite the strengths, several limitations need to be addressed in the current study. First, the association between tooth loss and cognitive impairment may be bidirectional because of the cross-sectional nature of the data and analysis. Although we used a time variable, poor cognitive function in AD patients can trigger periodontitis and poor oral hygiene, which ultimately leads to tooth loss due to incapability to clean their teeth and follow oral hygiene cautions or to regularly visit the dentist for specialized care [32]. Second, there are concerns concerning the reliability of AD diagnosis, which may be somewhat compromised because the diagnosis of AD depends on the disease codes in the database. However, as described in our previous work, we demonstrated that the coding of AD from the National Health Insurance Service Health Screening Cohort data had suitable validity for clinical AD diagnosis [33]. Third, most studies reporting these associations have considered that periodontitis is one of the major sources of tooth loss, which contributes to the enhancement of neuroinflammatory progression in the brain and ultimately results in cognitive impairment [34]. In addition to periodontitis, various oral diseases and conditions causing tooth loss, such as dental caries, soft tissue pathology, denture-related problems, and decreased denture use, were not considered in this study [35]. Fourth, we calculated the OR per 16 missing teeth to estimate the risk of AD associated with tooth loss in this study. Because the effect size of the loss of one tooth on AD occurrence was too small in our analysis, we chose to include the loss of 16 teeth instead of one tooth as our independent variable. Fifth, although potential confounding variables were adjusted in the analyses, there may have been a few unmeasured confounders, such as genetic risk factors, physical activity, and depression. Although we adjusted for variable factors in this study, general health status or social activity status could affect both tooth loss and AD.

## 5. Conclusions

In summary, our conclusions have implied that individuals with a higher number of missing teeth can have an increased risk of AD in the Korean population. This finding emphasizes the clinical importance of dental care and treatment for the prevention of the occurrence of AD later in life.

## Figures and Tables

**Figure 1 jcm-10-03763-f001:**
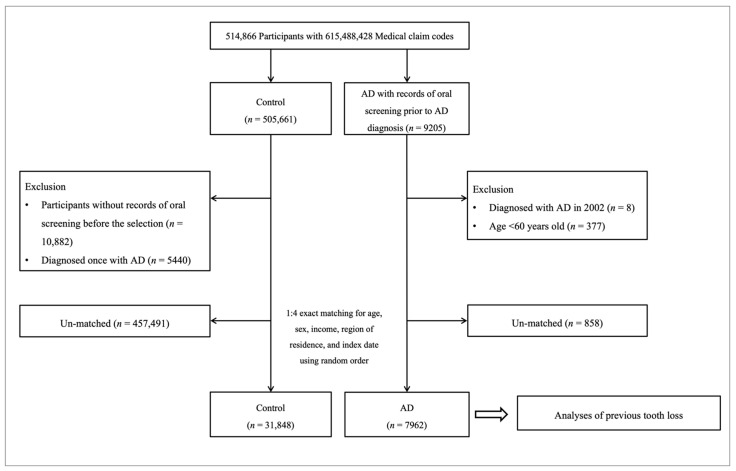
The current pathway of participant selection and inclusion in this study. Of the total 514,866 participants, 7962 Alzheimer’s disease participants were 1:4 matched with 31,848 control participants for age, sex, income, and region of residence.

**Figure 2 jcm-10-03763-f002:**
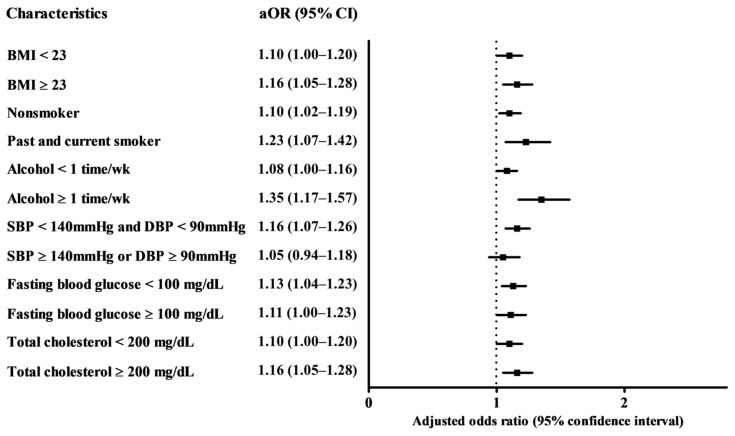
Odds ratios (95% confidence interval) of total tooth loss for Alzheimer’s disease in stratified subgroup according to obesity, smoking, alcohol consumption, blood pressure, fasting blood glucose, and total cholesterol.

**Figure 3 jcm-10-03763-f003:**
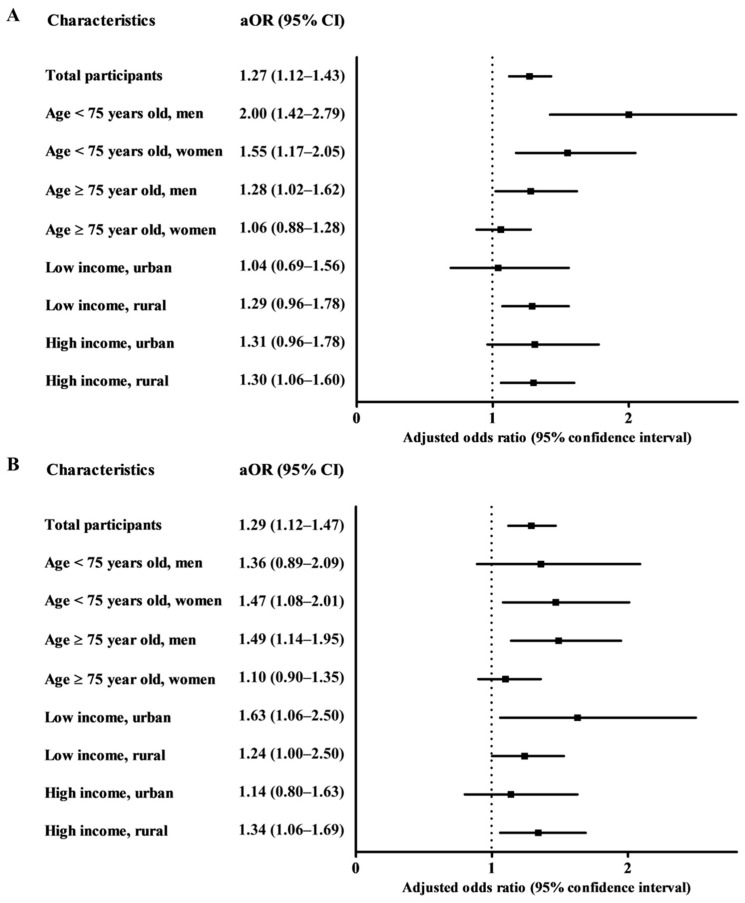
**(****A**) Odds ratios (95% confidence interval) of upper tooth loss for Alzheimer’s disease with subgroup analyses according to age and sex, income and region of residence. (**B**) Odds ratios (95% confidence interval) of lower tooth loss for Alzheimer’s disease with subgroup analyses according to age and sex, income and region of residence.

**Table 1 jcm-10-03763-t001:** General characteristics of participants.

Characteristics	Total Participants		
	AD	Control	*p*-Value
Age (years old, *n*, %)			1.000
60–64	513 (6.4)	2052 (6.4)	
65–69	1144 (14.4)	4576 (14.4)	
70–74	2074 (26.1)	8296 (26.1)	
75–79	2430 (30.5)	9720 (30.5)	
80–84	1571 (19.7)	6284 (19.7)	
85+	230 (2.9)	920 (2.9)	
Sex (*n*, %)			1.000
Male	3327 (41.8)	13,308 (41.8)	
Female	4635 (58.2)	18,540 (58.2)	
Income (*n*, %)			1.000
1 (lowest)	1543 (19.4)	6172 (19.4)	
2	904 (11.4)	3616 (11.4)	
3	1062 (13.3)	4248 (13.3)	
4	1418 (17.8)	5672 (17.8)	
5 (highest)	3035 (38.1)	12,140 (38.1)	
Region of residence (*n*, %)			1.000
Urban	2992 (37.6)	11,968 (37.6)	
Rural	4970 (62.4)	19,880 (62.4)	
Obesity (*n*, %)^3^			
Underweight	397 (5.0)	1345 (4.2)	<0.001 ^1^
Normal	3169 (39.8)	11,643 (36.6)	
Overweight	1950 (24.5)	8198 (25.7)	
Obese I	2233 (28.1)	9660 (30.3)	
Obese II	213 (2.7)	1002 (3.2)	
Smoking status (*n*, %)			<0.001 ^1^
Nonsmoker	6286 (79.0)	25,030 (78.6)	
Past smoker	859 (10.8)	3844 (12.1)	
Current smoker	817 (10.3)	2974 (9.3)	
Alcohol consumption (*n*, %)			<0.001 ^1^
<1 time a week	6058 (76.1)	23,100 (72.5)	
≥1 time a week	1904 (23.9)	8748 (27.5)	
Systolic blood pressure (*n*, %)			0.005 ^1^
<120 mmHg	1905 (23.9)	7214 (22.7)	
120–139 mmHg	3779 (47.5)	15,736 (49.4)	
≥140 mmHg	2278 (28.6)	8898 (27.9)	
Diastolic blood pressure (*n*, %)			0.023 ^1^
<80 mmHg	3699 (46.5)	15,188 (47.7)	
80–89 mmHg	2821 (35.4)	11,284 (35.4)	
≥90 mmHg	1442 (18.1)	5376 (16.9)	
Fasting blood glucose (*n*, %)			<0.001 ^1^
<100 mg/dL	4298 (54.0)	18,529 (58.2)	
100–125 mg/dL	2509 (31.5)	9952 (31.3)	
≥126 mg/dL	1155 (14.5)	3367 (10.6)	
Total cholesterol (*n*, %)			0.009 ^1^
<200 mg/dL	4421 (55.5)	17,878 (56.1)	
200–239 mg/dL	2432 (30.6)	9940 (31.2)	
≥240 mg/dL	1109 (13.9)	4030 (12.7)	
CCI score (score, *n*, %)			<0.001 ^1^
0	3000 (37.7)	18,232 (57.3)	
1	1902 (23.9)	5944 (18.7)	
2	1152 (14.5)	3359 (10.6)	
3	873 (11.0)	1964 (6.2)	
≥4	1035 (13.0)	2349 (7.4)	
Total tooth loss (mean, SD)	2.94 (6.09)	2.59 (5.6)	<0.001 ^2^
Total tooth loss (*n*, %)			<0.001 ^1^
0	4691 (58.9)	19,292 (60.6)	
1–2	1197 (15.0)	4916 (15.4)	
≥3	2074 (26.1)	7640 (24.0)	
Loss of upper teeth (mean, SD)	1.54 (3.44)	1.35 (3.16)	<0.001 ^2^
Loss of upper teeth (*n*, %)			<0.001 ^1^
0	5547 (69.7)	22,760 (71.5)	
1–2	1060 (13.3)	4206 (13.2)	
≥3	1355 (17.0)	4882 (15.3)	
Loss of lower teeth (mean, SD)	1.40 (3.03)	1.24 (2.81)	<0.001 ^2^
Loss of lower teeth (*n*, %)			<0.001 ^1^
0	5424 (68.1)	22,117 (69.5)	
1–2	1178 (14.8)	4954 (15.6)	
≥3	1360 (17.1)	4777 (15.0)	

Note: CCI, Charlson Comorbidity Index; SD, Standard deviation. ^1^ Chi-square test, significance at *p* < 0.05. ^2^ Independent *t* test. Significance at *p* < 0.05. ^3^ Obesity (BMI, body mass index, kg/m^2^) was categorized as <18.5 (underweight), ≥18.5 to <23 (normal), ≥23 to <25 (overweight), ≥25 to <30 (obese I), and ≥30 (obese II).

**Table 2 jcm-10-03763-t002:** Odds ratios for Alzheimer’s disease per 16 missing teeth with subgroups stratified by age, sex, income, and region of residence.

Characteristics	Odds Ratios for AD (95% Confidence Interval)			
	Crude1 ^2^	*p*-Value	Adjusted ^2,3^	*p*-Value
Total participants (*n* = 39,810)				
Total tooth loss	1.19 (1.11–1.27)	<0.001 ^1^	1.15 (1.07–1.23)	<0.001 ^1^
Age <75 years old, men (*n* = 7895)				
Total tooth loss	1.45 (1.19–1.78)	<0.001 ^1^	1.32 (1.07–1.64)	0.010 ^1^
Age <75 years old, women (*n* = 10,760)				
Total tooth loss	1.32 (1.13–1.54)	<0.001 ^1^	1.27 (1.09–1.49)	0.003 ^1^
Age ≥75 years old, men (*n* = 8740)				
Total tooth loss	1.24 (1.08–1.41)	0.002 ^1^	1.20 (1.05–1.37)	0.008 ^1^
Age ≥75 years old, women (*n* = 12,415)				
Total tooth loss	1.43 (0.92–2.21)	0.109	1.40 (0.90–2.17)	0.138
Low income, urban (*n* = 5720)				
Total tooth loss	1.20 (0.96–1.50)	0.101	1.16 (0.92–1.45)	0.212
Low income, rural (*n* = 11,825)				
Total tooth loss	1.18 (1.06-1.31)	0.002 ^1^	1.14 (1.03–1.27)	0.014 ^1^
High income, urban (*n* = 9240)				
Total tooth loss	1.20 (1.01–1.43)	0.039 ^1^	1.13 (0.95–1.35)	0.179
High income, rural (*n* = 13,025)				
Total tooth loss	1.20 (1.07–1.34)	0.002 ^1^	1.17 (1.04–1.31)	0.009 ^1^

Note: CCI, Charlson Comorbidity Index. ^1^ Conditional logistic regression model, significance at *p* < 0.05. ^2^ Models stratified by age, sex, income, and region of residence. ^3^ Models adjusted for obesity, smoking, alcohol consumption, systolic blood pressure, diastolic blood pressure, fasting blood glucose, total cholesterol, and CCI scores.

## Data Availability

The current article used a national sample cohort and does not involve data that can be available.

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
