# Peer review of "Association between Tooth Loss and Alzheimer’s Disease in a Nested Case–Control Study Based on a National Health Screening Cohort"

_jcm, 2021, doi:10.3390/jcm10173763_

Round 1

Reviewer 1 Report

The authors present an interesting work associating the loss of teeth to Alzheimer's disease. The work is well written and nicely presented, and I support publication. It will be great if the following points can be taken into account. 

(1) In lines 48-50, the authors mentioned that numerous publications have focused on the relationship between tooth loss and dementia, inferring that tooth loss can be an amendable risk factor for dementia. It would be great if the authors could include more references and elaborate on what are the observations globally related to tooth loss and dementia, especially indicating if there were any regional differences observed previously. This will help to support that a study based in Korea can be generally applicable to the understanding of this field of topic worldwide.

  (2) In lines 177-180, the authors have briefly discussed other factors such as obesity, smoking status, alcohol consumption, BPs, fasting glucose and cholesterol. It was not clear why OR was higher for the different subgroups except alcohol consumption less once a week and SBP ≥140 mmHg and DBP ≥90 mmHg. Aren't the following also less than 1.15 (e.g. BMI<23, non-smoker, fasting glucose and total cholesterol < 200mg/dL). Maybe it's good to elaborate and provide a clear discussion on this.      (3) In lines 244-246, the authors stated that the strengths of this work are that it was a large population-based analysis with nationwide representativeness of all diagnosed AD cases without exceptions, which ensures statistical reliability and power. It might be too strong to say "nationwide representativeness of all diagnosed AD cases without exceptions" since there were multiple AD patients excluded in this study (e.g. 5440 who were diagnosed with AD once).   (4) In lines 264-267, the author mentioned "In addition to periodontitis, various oral diseases and conditions causing tooth loss, such as dental caries, soft tissue pathology, denture-related problems, and decreased denture use, were not considered in this study". This is an important limitation of the study which needs to be addressed in future work. However, I am wondering if the authors have taken into account the loss of teeth due to oral related diseases only vs physical loss of teeth (e.g. loss of teeth due to impact such as accident). It makes more sense to relate loss of teeth due to oral diseases to AD as compared to accidental loss of teeth. This is important to support the authors' conclusion: "This finding emphasizes the clinical importance of dental care and treatment for the prevention of the occurrence of AD later in life." (Lines 275-277)

Reviewer 2 Report

This is the survey to explore the association between tooth loss and AD in a large sample of participants in Korea.  The authors concluded that individuals  with a higher number of missing teeth can have an increased risk of AD.

  • In the introduction, it is stated suddenly that  tooth loss in the upper and lower jaws are analyzed separately.  Why the authors need to analyze upper and lower separately? More info please.
  • In the section of 2.3. (Line 81-84), the number of tooth loss is only counted during 2002 through 2015, or total missing teeth ever? According to results, it seems the number of tooth loss would be only counted during 2002 through 2015, but it is unclear to me. Explain the difference between tooth loss and missing teeth in your study.
  • Line 86, all of subjects of this study have AD? This sentence should be rewriting.
  • Line 92. Why AD participants below the age of 60 yrs were excluded? I feel n=377 is so many and not negligible.  
  • Line 110, how authors classified income group is unclear. Please specify the amount of income as described previous study. 
  • Line 134-136, the author reported the ORs of AD participants compared to the control participants per 16 missing teeth rather than one missing tooth. I understand that the effect size of just one missing tooth was small, but I think this method seems to be unconventional. Do you have any reference using similar method?
  • Line 158, what is SCD?
  • Most of discussion section occupy references and the authors did not address to discuss about their results. Discussion should be rewrite.
  • I would say to recommend authors following STROBE guidelines for reporting of this type of study.
  • In my opinion, to prevent tooth loss, regular dental care visit seems to be important. Also regular dental visit may be influenced by income. Is it uncommon to add dental visit in variables?

Reviewer 3 Report

The strength of this manuscript is that the authors were able to draw upon a large national health survey from South Korea with data on cognitive function to match subjects with similar demographic data in the major categories of age, gender, educational level, and  income.  and use the data from total tooth counts as they are associated with lower cognitive function with appropriate regression analysis.  With this large data set, the authors have presented strong confirmatory evidence of the association between tooth loss and cognitive decline.  Therefore, the manuscript does add to the epidemiological literature in this important area of clinical medicine and dentistry

The major concern from this reviewer is that while the results, and discussion section are clearly written and the conclusions that an association between tooth loss and decline in cognitive section are justified, there are several major and minor concerns to be addressed in the abstract, introduction and methods section.   

  1. The statement in the introduction that the literature shows that tooth loss LEADS to AD has not been demonstrated in the literature. This would be a cause-and effect relationship.   Rather the literature only shows an association.  This difference between an association and a cause-and-effect relationship should be clarified throughout the manuscript, as it is in the discussion section.
  2. A justification of comparing the association of tooth loss by arch and AD is not given.
  3. In the methods section 2.2 more detail is needed on how a diagnosis of AD was determined
  4. IN the methods section 2.4 the term ”washout period” for the diagnosis of AD needs to be explained
  5. The introduction and abstract sections have numerous grammatical errors which require review and corrections by an English editor.

Round 2

Reviewer 2 Report

  • Line 116-117, to avoid arbitrary categorization, authors should state how groups were categorized into 5 groups. 'Lowest income to highest income' is not enough. Did you use quintile to categorize group? If so, just add the sentence. If not, add how author categorized group concretely, for example, monthly income <1,000 USD, 1,001-2,000 USD, 2,001-3,000 USD, 3,001-4,000 USD, >4,000 USD. 
  • Reference 10 and 29 are same.
